# Early diagnosis of dengue: Diagnostic utility of the SD BIOLINE Dengue Duo rapid test in Reunion Island

Olivier Maillard[1,2☯]*, Jeanne Belot[3☯], Thibault Adenis[3], Olivier Rollot[1,2], Antoine Adenis[4,5], Bertrand Guihard[6], Patrick Gérardin[1,2], Antoine Bertolotti[2,7]

1 Department of Public Health and Research, CHU Réunion, Saint-Pierre, Reunion, France, 2 Clinical Investigation Center, INSERM CIC1410, CHU Réunion, Saint Pierre, Reunion, France, 3 Department of Emergency Medicine, CHU Réunion, Saint-Pierre, Reunion, France, 4 Department of Public Health and Research, CH Andrée Rosemon, Cayenne, French Guiana, France, 5 Clinical Investigation Center, INSERM CIC1424, CH Andrée Rosemon, Cayenne, French Guiana, France, 6 Department of Emergency Medicine, CHU Réunion, Saint Denis, Reunion, France, 7 Department of Infectious Diseases and Dermatology, CHU Réunion, Saint-Pierre, Reunion, France

☯ These authors contributed equally to this work.
* olmaillard@yahoo.fr

**Data Availability Statement:** All relevant data are within the manuscript and its Supporting Information files. The data underlying the results

## Abstract

### Background

In Reunion Island, dengue outbreaks have been occurring since 2018. The healthcare facilities are facing the problem of managing a massive influx of patients and a growing care burden. The aim of this study was to evaluate the performance of the SD Bioline Dengue Duo rapid diagnostic test in adults consulting at an emergency department during the 2019 epidemic.

### Methodology/Principal findings

This retrospective study of diagnostic accuracy included patients over 18 years old, suspected of dengue, who were admitted to emergency units of the University Hospital of Reunion between the 1st of January and 30th of June, 2019, and were tested for dengue fever with the SD Bioline Dengue Duo rapid diagnostic test and reverse transcriptase polymerase chain reaction. Over the study period, 2099 patients were screened retrospectively. Of them, 671 patients matched the inclusion criteria. The overall rapid diagnostic test performance was 42% for sensitivity and 15% for specificity. The non-structural 1 antigen component had a good specificity of 82% but a low sensitivity of 12%. The immunoglobulin M component had a sensitivity of 28% and a specificity of 33%. Sensitivities were slightly improved beyond the 5th day of illness compared to the early stage for all components, but only the non-structural 1 antigen component had a better specificity of 91%. Furthermore, predictive values were low and post-test probabilities never improved pre-test probabilities in our setting.

presented in the study are available from the INSERM CIC1410 (cic@chu-reunion.fr).

**Funding:** The author(s) received no specific funding for this work.

**Competing interests:** The authors have declared that no competing interests exist.

## Conclusions/Significance

These results suggest that the SD Bioline Dengue Duo RDT did not achieve sufficient performance levels to rule in, or discard, an early point of care dengue diagnosis in the emergency department during the 2019 epidemic in Reunion.

## Author summary

Dengue fever is the most common mosquito-borne viral disease. Most often mild, it can progress to a severe form that can lead to death. Since 2018, dengue outbreaks have occurred in Reunion Island, a French overseas territory, with an increasing number of confirmed cases and related deaths reported over these years. The care burden of dengue during epidemics often exceeds the capacity of health care facilities. To optimize the management of cases, it is necessary to diagnose infected patients early. However, clinical diagnosis is difficult as dengue occurs as an influenza-like illness with broad spectrum and non-specific symptoms, and laboratory confirmation is expensive, not immediate and not always available. Therefore, rapid diagnostic tests (RDT) have been developed and could serve as a sensitive, specific, robust point of care diagnostic tool. Although promising, RDT performance is variable depending on the setting. In this study, we evaluated the performance of the SD Bioline Dengue Duo RDT in emergency departments of Reunion during the 2019 epidemic. Results suggest that the SD Bioline Dengue Duo RDT did not achieve sufficient performance levels to rule in or discard an early point of care dengue diagnosis in our setting.

## Introduction

Dengue fever is the most common mosquito-borne viral disease [1]. The massive increase of dengue incidence, with 400 million estimated annual cases worldwide combined with the geographical extension of its vector, the *Aedes* mosquito, makes dengue a major public health concern [1,2]. Previously known as a tropical and subtropical disease, dengue fever is now considered as a seasonal epidemic risk in Europe [3].

Dengue outbreaks have been growing in Reunion island since 2018, with an increasing number of confirmed cases and related deaths reported [4,5]. The care burden of dengue during epidemics often exceeds the capacity of health care facilities. To optimize the management of cases, it is necessary to diagnose infected patients early. However, clinical diagnosis is difficult as dengue occurs as an influenza-like illness with broad spectrum and non-specific symptoms. Biological diagnosis' confirmation may be achieved by either virus isolation, or molecular amplification of dengue virus (DENV) RNA with reverse transcriptase polymerase chain reaction (RT-PCR), or immunoassays to detect DENV non-structural 1 (NS1) antigen alone or in combination with DENV IgM and IgG antibodies. All of these techniques are expensive, not immediate and not always available [6–8]. To meet the need for early and accurate diagnosis, rapid diagnostic tests (RDT) have been developed [7,9], which can provide results within fifteen minutes. They could serve as a sensitive, specific, robust point of care diagnostic tool and do not require any equipment [7]. Although promising, RDT sensibility and specificity vary depending on the stage of the disease and the serotype, and both decrease in case of secondary infection [7–11]. The aim of this retrospective study was to assess the performance of the SD Bioline Dengue Duo RDT (Standard Diagnostics, South Korea) in

Reunion Island during the 2019 epidemic and whether physicians could make a diagnosis on the basis of its results in this setting.

## Methods

### Ethical approval

This monocentric, observational, retrospective, diagnostic study was conducted according to the MR-004 reference methodology from the National Commission of Informatics and Liberties (CNIL), which complies with the General Data Protection Regulation. In accordance with French regulations, this retrospective study did not require approval from an ethics committee. It was reported according to the STARD (Standard for Reporting of Diagnostic Accuracy) guideline (S1 Table). The EPIDENGUE database was registered in the national health data hub (n˚ F20201021104344). Non-refusal of participation was collected. Data was treated anonymously from patients' medical records.

### Study design

Data was retrieved from the EPIDENGUE database, which is a retrospective collection from medical files of the characteristics of consecutive patients suspected of dengue fever, who were admitted to emergency departments of the University Hospital of Reunion between 1st of January and 30th of June, 2019. The EPIDENGUE database was set up to study dengue fever in Reunionese patients first during 2019 epidemic while a prospective study was launched at the same time but with limited recruitment because emergency services were overwhelmed during the epidemic. A retrospective study of diagnostic accuracy of SD Bioline dengue duo RDT was conducted in patients over 18 years of age because only few children had both RDT and RT-PCR. Of them, we searched for all patients who underwent RDT for dengue diagnosis at admission. Patients suspected of having dengue fever underwent point-of-care testing by blood sampling on admission. All had an RDT evaluated by a primary care physician, and a few also had a reference test, i.e. real-time RT-PCR during the acute phase (from day 0 to day 5), RT-PCR and IgM serology (from day 5 to day 7), and IgM serology during the convalescent phase (from day 5 to day 10), performed by laboratory staff blinded to RDT results. The serology wasn't systematically double checked three weeks later during the epidemic, or in ambulatory care, so that data link could not be made with confidence. This is the reason why only cases with both RDT and RT-PCR results were included in the study.

The sample size derived from the availability of results of the RDT used at point-of-care during this epidemic and the RT-PCR used for on-site routine diagnosis over the period.

In this conditions, a sample size of 547 patients produces a two-sided 95% confidence interval with a width ranging from 0.078 to 0,127 for component sensitivities from 0.1 to 0.9, and from 0.072 and 0.117 for a component specificities from 0.1 to 0.9. Sample size calculation was performed using PASS software (PASS 2020, NCSS, LLC, Kaysville, Utah, USA).

### Diagnostic tests

The diagnostic test under study was the SD Bioline Dengue Duo (Standard Diagnostics, South Korea). This is a rapid immunochromatographic one-step assay to detect simultaneously the NS1 antigen and IgM and IgG DENV antibodies. The presence of colored lines in both result windows (control and patient) indicates a positive result. A faint line in NS1 or IgM antibodies was considered positive, in accordance with the manufacturer's instructions. The results of the IgG antibodies were not taken into consideration because they are not used to validate the diagnosis of acute infection in current practice. The reference laboratory test for dengue

diagnosis confirmation in the study was the Tropical Fever Core multiplex RT-PCR (Fast Track Diagnostics, Luxembourg), which was a conventional, two step, real-time RT-PCR used for on-site diagnosis at the time of the epidemic. This kit was chosen as routine, as it could detect dengue (without differentiation between serotypes), chikungunya and West Nile viruses, *Leptospira spp.*, *Rickettsia spp.* and *Salmonella spp.*, and *Plasmodium spp.*, which are pathogens that could circulate in the South West Indian Ocean region. In the case of PCR inhibitors, which could be detected by negative internal controls, results were rendered non interpretable or doubtful but not negative, and a new sample was required, or a second RNA extraction. All procedures were performed according to the manufacturer's protocols. Serology was not done routinely, but when there was a positive IgM RDT or a strong suspicion of dengue fever and a negative RT-PCR result.

## Final diagnosis

The patients were considered confirmed dengue cases if they had positive RT-PCR or positive NS1 RDT. Probable dengue cases were identified as defined by the World Health Organization (WHO) in 2009, without positive biological diagnostic tests but with clinical and biological features compatible with dengue. Otherwise, patients were considered non-probable dengue cases.

## Statistical analysis

Categorical variables were summarized with numbers and percentages, and comparisons between groups were performed using Chi-2 or Fisher's exact test. Continuous variables were described as means and standard deviations or medians and interquartile ranges. Multiple comparisons were performed using analyses of variance (ANOVA) or Kruskall-Wallis test, as appropriate, and the Mood's median test for comparison of medians. The normality of the distributions was checked by the Shapiro-Wilk test and the homogeneity of variances by the Levene test.

Intrinsic characteristics of the RDT were reported in terms of sensitivity, specificity, and positive and negative likelihood ratios with their 95% confidence intervals. Subgroup analyses by diagnosis (lab-confirmed vs probable dengue and non-probable dengue) and duration from illness onset (DIO) were also performed. To assess diagnostic utility, predictive values and post-test probabilities (95%CI) of dengue for positive and negative tests were calculated for the whole RDT and components, if information on component results was available in patients' files. Pre-test probability (prevalence) of dengue among participants was first converted to odds equal to prevalence/(1-prevalence). Pre-test odds was then multiplied with corresponding LR to get post-test odds that were converted back to probabilities equal to odds/(1 + odds) [12].

Statistical analyses were performed using SPSS software (IBM SPSS 23.0, Amonk, NY, USA). All tests were two-tailed and a P-value below 0.05 was considered as statistically significant.

## Results

Of the 2,099 patients over 18 years of age who were admitted in emergency departments of the University Hospital of Reunion during the first semester of 2019, 946 patients were sampled for RDT. Among them, 671 patients also underwent RT-PCR, 286 (43%) were RT-PCR positive, 449 (67%) were RDT positive, of which 85 (19%) were NS1 antigen positive, 267 (59%) were IgM antibodies positive, 88 (20%) were IgG antibodies positive, and 124 (28%) were globally positive without other details in their medical files (Fig 1). Among the 547 patients with

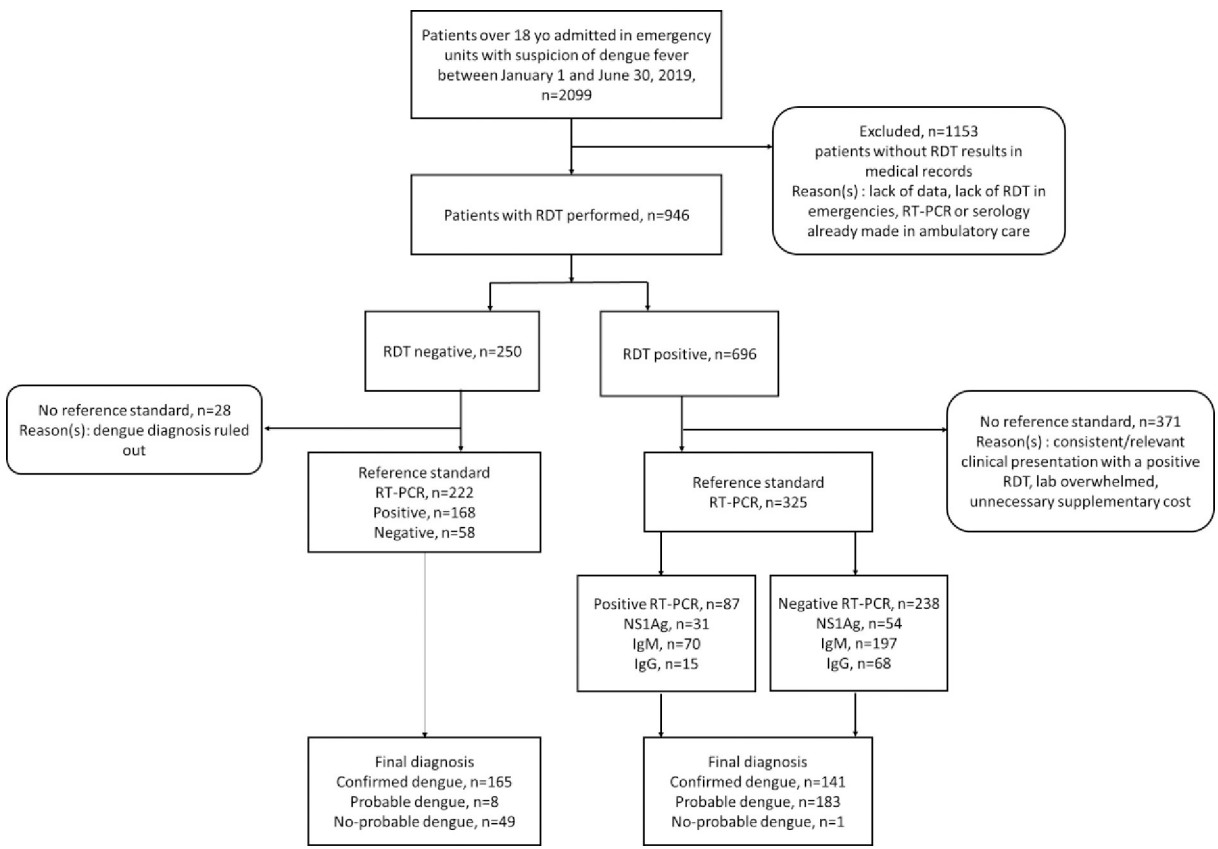

**Fig 1. Flowchart of sample selection.** Legend: RDT: rapid diagnostic test; RT-PCR: reverse transcriptase polymerase chain reaction; NS1 Ag: non-structural 1 antigen; IgM: immunoglobulin M; IgG: immunoglobulin G.

detailed RDT results, dengue diagnosis was confirmed in 306 (56%). It was probable in 191 (35%) and non-probable in 50 (9%).

The demographic and clinical features as well as biological data of included cases are reported in Table 1. Confirmed or probable dengue patients were older than non-probable dengue patients (median age: 55 or 60 versus 41 years old, $P$ = 0.032). Fever was the main symptom reported in 288 (94%) confirmed cases. The other most common symptoms reported in confirmed cases were asthenia in 213 (70%), myalgia in 194 (63%), gastrointestinal symptoms in 184 (60%), headache in 167 (55%), anorexia in 164 (54%), arthralgia in 143 (47%), cutaneous signs in 109 (36%) and neurological disorders in 85 (28%) with statistical differences between groups ($P<0.001$). They were less frequent in other groups. White blood cells, neutrophils, lymphocytes, platelets, and prothrombin ratio (PR) were significantly different between the groups. They were lower in confirmed cases than other groups except PR, which was lower in non-probable dengue patients.

Intrinsic properties of the SD Bioline Dengue Duo RDT are given in Table 2. The overall sensitivity and specificity were 42% (95%CI: 37–48) and 15% (95%CI:11–19). The NS1 component had a sensitivity of 12% (95%CI: 9–17) and specificity of 82% (95%CI: 77–86). The IgM component had a sensitivity of 28% (95%CI: 22–34) and a specificity of 33% (95%CI: 28–39). The overall sensitivity (NS1 or IgM component) was 33% (95%CI: 27–39) and the overall specificity was 24% (95%CI: 20–30). The best specificity, 91% (95%CI: 87–94), was found for the combined NS1 and IgM components with a sensitivity of 8% (95%CI: 5–12).

**Table 1. Characteristics of study population, Reunion, 2019 (N = 671).**

| Characteristics | Confirmed dengue (n = 306) | Probable dengue (n = 191) | Non-probable dengue (n = 50) | P value |
|---|---|---|---|---|
| **Demographic** | | | | |
| Male | 157 (51) | 91 (48) | 25 (50) | 0.729 |
| Female | 149 (49) | 100 (52) | 25 (50) | |
| Age median(range) | 55 (18–96) | 60 (19–91) | 41 (18–96) | 0.032 |
| **Background** | | | | |
| Charlson score mean±SD | 2.4 ± 2.8 (306) | 2.6 ± 2.8 (191) | 1.4 ± 2.1 (50) | 0.006 |
| 0 | 109 (36) | 63 (33) | 29 (58) | |
| 1 | 47 (15) | 20 (11) | 6 (12) | |
| 2 | 32 (11) | 26 (14) | 2 (4) | |
| 3 | 36 (12) | 29 (15) | 5 (10) | |
| 4 | 21 (7) | 15 (8) | 3 (6) | |
| 5 | 19 (6) | 11 (6) | 2 (4) | |
| < 5 | 244 (80) | 153 (80) | 45 (90) | 0.177 |
| ≥ 5 | 62 (20) | 38 (20) | 5 (10) | |
| **Clinical features** | | | | |
| Fever | 288 (94) | 121 (63) | 9 (18) | <0.001 |
| Asthenia | 213 (70) | 123 (64) | 6 (12) | <0.001 |
| Anorexia | 164 (54) | 83 (44) | 4 (8) | <0.001 |
| Headache | 167 (55) | 72 (38) | 6 (12) | <0.001 |
| Retro orbital pain | 65 (21) | 24 (13) | 4 (8) | <0.001 |
| Myalgia | 194 (63) | 79 (41) | 6 (12) | <0.001 |
| Arthralgia | 143 (47) | 56 (29) | 4 (8) | <0.001 |
| Backache | 68 (22) | 27 (14) | 1 (2) | <0.001 |
| Cutaneous signs[†] | 109 (36) | 29 (15) | 1 (2) | <0.001 |
| GI symptoms[‡] | 184 (60) | 88 (46) | 3 (6) | <0.001 |
| Neurological disorders | 85 (28) | 35 (18) | 0 | <0.001 |
| Bleeding* | 48 (16) | 15 (8) | 1 (2) | 0.001 |
| Severe dengue** | 68 (22) | 42 (22) | - | 0.376 |
| DIO (days) | 2.2 ± 2.3 (304) | 3.3 ± 3.9 (162) | 3.8 ± 4.9 (15) | 0.003 |
| ≤ 5 days | 275 (90) | 129 (80) | 12 (80) | 0.002 |
| > 5 days | 29 (10) | 33 (20) | 3 (20) | |
| **Hospitalization** | 118 (39) | 60 (31) | 4 (8) | <0.001 |
| LOS (days) | 5.7 ± 3.8 (116) | 6.3 ± 4.8 (59) | 14.3 ± 12.1 (4) | 0.168 |
| **Biological parameters** | | | | |
| Hemoglobin (g/dL) | 13.7 ± 2.0 (260) | 13.5 ± 1.9 (146) | 13.8 ± 1.9 (17) | 0.604 |
| Hematocrit (%) | 40.1 ± 5.1 (260) | 39.8 ± 4.8 (146) | 40.9 ± 4.4 (17) | 0.513 |
| WBC (G/L) | 5.3 ± 4.0 (260) | 7.2 ± 4.0 (146) | 11.5 ± 7.8 (17) | <0.001 |
| Neutrophils (G/L) | 3.8 ± 2.7 (253) | 5.1 ± 3.7 (146) | 8.8 ± 7.9 (16) | <0.001 |
| Lymphocyts (G/L) | 0.8 ± 2.1 (253) | 1.3 ± 0.8 (146) | 2.1 ± 0.8 (16) | <0.001 |
| Platelets (G/L) | 166 ± 77 (260) | 192 ± 87 (146) | 203 ± 60 (17) | 0.009 |
| PR (%) | 89 ±16 (251) | 91 ± 20 (144) | 75 ± 27 (16) | 0.002 |

[‡] Gastrointestinal symptoms: abdominal pain, nausea, vomiting, diarrhea

[†] Cutaneous signs: conjunctivitis, dysgeusia, oral damage, erythema (located or diffuse), itching

Bleeding: gingivorragia, menometrorragia, hemoptysis, epistaxis, gastrointestinal bleeding, purpura

**Severe dengue: OMS definition 2009

DIO: duration from illness onset; LOS: length of stay; WBC: white blood cells; PR: prothrombin ratio

**Table 2. Performance of SD Bioline Dengue Duo rapid diagnostic test, Reunion, 2019 (N = 671).**

| RDT components | Sensitivity (%) | Specificity (%) | PLR | NLR |
|---|---|---|---|---|
| **Overall** | 42 (37–48) | 15 (11–19) | 0.50 (0.43–0.57) | 3.90 (3.01–5.05) |
| **NS1 Ag** | 12 (9–17) | 82 (77–86) | 0.67 (0.45–1.01) | 1.07 (1.00–1.15) |
| **IgM** | 28 (22–34) | 33 (28–39) | 0.42 (0.34–0.52) | 2.17 (1.82–2.60) |
| **NS1 Ag or IgM** | 33 (27–39) | 24 (20–30) | 0.43 (0.36–0.52) | 2.76 (2.22–3.44) |
| **NS1 Ag and IgM** | 8 (5–12) | 91 (87–94) | 0.79 (0.45–1.39) | 1.02 (0.97–1.08) |

RDT: rapid diagnostic test; NS1 Ag: non-structural 1 antigen; IgM: immunoglobulin M; PLR: positive likelihood ratio; NLR: negative likelihood ratio.

Furthermore, performance of the RDT was analyzed according to duration from illness onset (DIO) at the point of admission to the emergency department. This evaluation was conducted in a subpopulation with available DIO (88% of total). More than 80% of patients visited hospital in the first five days of symptoms. The sensitivity and specificity of the RDT for the NS1 component and the sensitivity for the IgM component were improved in patients admitted after five days from illness onset, compared to within the first five days of illness onset. The detailed results are presented in S2 and S3 Tables. For information, performance of the IgG component is shown in S4 Table.

Diagnostic accuracy depends on the pre-test probability through the prevalence of the disease. We calculated predictive values and post-test probabilities of dengue for the RDT with a pre-test probability of dengue of 46%, which corresponded to the sampling conditions in our setting (Table 3). Post-test probability of dengue for at least one positive test component never improved the pre-test probability. Moreover, post-test probability of dengue for negative test components always exceeded the post-test probability of dengue for at least one positive test component. The conditions of sampling did not reflect the prevalence in our setting. Analyses in different pre-test probabilities ranging from 5% to 30% are detailed in S5 Table.

## Discussion

In this retrospective study, the comparative performance of the NS1 antigen and IgM antibodies components of the SD Bioline Dengue Duo RDT were analyzed relative to RT-PCR and by DIO. The NS1 and IgM antibodies components presented higher specificity when combined than separately. Conversely, the sensitivity was very low. Specificity of the IgM component

**Table 3. Diagnostic utility estimates and their 95% confidence interval for the RDT with a pre-test probability of dengue of 46% (sampling conditions), Reunion, 2019 (N = 671).**

| RDT components | PPV (%) | NPV (%) | Diagnostic accuracy (%) | Post-test probability of dengue for positive test component (%) | Post-test probability of dengue for negative test component (%) |
|---|---|---|---|---|---|
| **Overall** | 27 (24–30) | 26 (21–31) | 27 (23–30) | 27 (24–30) | 74 (69–79) |
| **NS1 Ag** | 36 (28–46) | 52 (50–54) | 50 (45–54) | 37 (28–46) | 48 (46–50) |
| **IgM** | 26 (22–31) | 35 (31–39) | 31 (27–35) | 26 (22–31) | 65 (61–69) |
| **NS1 Ag or IgM** | 27 (23–31) | 30 (25–35) | 28 (24–32) | 27 (23–31) | 70 (65–75) |
| **NS1 Ag and IgM** | 40 (28–54) | 53 (52–55) | 52 (48–57) | 40 (28–54) | 47 (45–48) |

RDT: rapid diagnostic test; NS1 Ag: non-structural 1 antigen; IgM: immunoglobulin M; PPV: positive predictive value; NPV: negative predictive value.

alone was exceptionally low either in acute or convalescent phase. There was an improvement of sensitivity of NS1 and IgM components when the DIO was higher than 5 days, but most patients (87%) consulted within the first five days of illness onset. Therefore an RDT for early diagnosis at point of care was needed in our setting.

Most previous studies on the SD Bioline Dengue Duo RDT found varying but better sensitivities, either for NS1 component ranging between 39–88% or for IgM component 14–98%, or for combined components 31–91%. They found better specificities for IgM component ranging between 80–96% [9–11,13,14]. Most of these studies were conducted on hospitalized patients or hospital serum samples. They usually have more severe disease presentation and consequently more inflammation and a greater immune response [14]. In this study, confirmed cases had a wide spectrum of symptoms going from asymptomatic to severe dengue, but most had mild symptoms. Moreover, all patients included were suspected of dengue and had point of care testing, whereas many studies were conducted on a convenient serum sample collection [10,11,13,15–18].

The choice of the reference diagnostic test used in performance RDT studies is known to influence the results [14,19]. In the study of Kikuti et al., the overall sensitivity for the NS1 RDT was 38,6%, but when the NS1 ELISA was the only reference diagnostic test used, the RDT achieved a high sensitivity (90,4%) [14]. The majority of studies use ELISA tests as a reference diagnostic test, or a combined strategy with ELISA and RT-PCR [10,11,13,14].

The persistence of residual NS1 antigen in patients' blood during the convalescent phase while DENV viremia quickly declines may have decreased RT PCR sensitivity and subsequently NS1 RDT specificity [1].

IgM antibodies appear with a later delay in primary infections (>6 days) than in secondary infections (day 4). IgM antibodies have a longer duration for primary infection (until three months) than for secondary infection (<10 days). Moreover, there are cross reactions with other infectious diseases [10,20] which is well-known with other flaviviruses, but also chikungunya, of which a large epidemic occurred in Reunion in 2005–2006 [21,22]. It is also reported with leptospirosis and rickettsioses, which are endemic in Reunion, with overlapping features at the acute phase with dengue [23,24]. It could explain, in our setting of primary dengue infections, the low sensitivity and specificity of IgM RDT when compared to RT PCR as reference test.

When focusing on the 89 positive IgM RDT at Day 0 and Day 1 of the onset of illness, only 27 (30%) were RT-PCR positive and 67 (75%) had a serology test but only five (6%) were positive in IgM antibodies. As they were also positive in IgG antibodies, these five patients seemed to be in the convalescent phase of a previous dengue fever. However, three also had a positive RT-PCR that could be explained by a secondary dengue, but serum neutralization was not routinely done in 2019 because Reunionese people were presumed to be naïve to dengue. Even if some patients may have mistaken the date of onset of symptoms, these results also illustrate the bad performance of IgM RDT component compared to IgM ELISA, due to a high rate of false positives at Day 0 and Day 1.

In our study, the DENV infecting serotype was not determined routinely but in a sample of patients, and in 2019 the main serotype reported was DENV-2. Recent studies from Asia and Venezuela have found that the sensitivity of NS1 RDT would be lower for DENV-2 infections [25,26].

This study may suffer from selection bias because of the retrospective design, the hospital's recruitment and the lack of information in the patients' medical files (124 RDT results were reported as positive without information on the components results and some DIO could not have been estimated). Furthermore, the reference diagnostic test was only RT-PCR because most patients consulted within the first five days of illness, thus a few had a serology done at

the time of RDT, with often only one IgM ELISA result in their medical file, and a positive result might represent either an active or a previous, recent infection.

In conclusion, our findings suggest that the SD Bioline Dengue Duo RDT does not achieve sufficient performance levels to rule in or discard a dengue diagnosis at point of care in our setting. Dengue endemization with yearly epidemics in Reunion Island challenges the diagnosis of dengue fever in the context of other endemic diseases with the same clinical presentation at an early stage of management.

## Supporting information

**S1 Table. STARD (Standard for Reporting of Diagnostic Accuracy) 2015 checklist.** (DOCX)

**S2 Table. Performance of RDT with duration from illness onset ≤ 5 days vs > 5 days, Reunion, 2019 (N = 671). Legend:** RDT: rapid diagnostic test; NS1 Ag: non-structural 1 antigen; IgM: immunoglobulin M; PLR: positive likelihood ratio; NLR: negative likelihood ratio. (DOCX)

**S3 Table. Performance of NS1 and IgM RDT depending on the duration from illness onset, Reunion, 2019 (N = 547). Legend:** RDT: rapid diagnostic test; NS1 Ag: non-structural 1 antigen; IgM: immunoglobulin M; PLR: positive likelihood ratio; NLR: negative likelihood ratio. (DOCX)

**S4 Table. Performance of IgG RDT with duration from illness onset ≤ 5 days vs > 5 days, Reunion, 2019 (N = 547). Legend:** RDT: rapid diagnostic test; IgG: immunoglobulin G; PLR: positive likelihood ratio; NLR: negative likelihood ratio. (DOCX)

**S5 Table. Post-test probabilities and their 95% confidence interval for the RDT according to pre-test probability of dengue, Reunion, 2019 (N = 671). Legend:** CI: confidence interval. (DOCX)

## Acknowledgments

All authors thank the emergency departments' staff of the University Hospital of Reunion for rigorously filling out the patient records and all collaborators of the EPIDENGUE project for filling out the case report form, as well as our copy editor Jennifer Sanders.

## Author Contributions

**Conceptualization:** Olivier Maillard, Jeanne Belot, Patrick Gérardin, Antoine Bertolotti.

**Data curation:** Olivier Maillard, Jeanne Belot, Thibault Adenis, Bertrand Guihard, Patrick Gérardin, Antoine Bertolotti.

**Formal analysis:** Olivier Maillard, Olivier Rollot.

**Investigation:** Jeanne Belot, Thibault Adenis, Bertrand Guihard, Patrick Gérardin, Antoine Bertolotti.

**Methodology:** Olivier Maillard, Patrick Gérardin, Antoine Bertolotti.

**Supervision:** Olivier Maillard, Patrick Gérardin, Antoine Bertolotti.

**Writing – original draft:** Olivier Maillard, Jeanne Belot.

**Writing – review & editing:** Thibault Adenis, Olivier Rollot, Antoine Adenis, Bertrand Guihard, Patrick Gérardin, Antoine Bertolotti.

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
