## [Decision Letter · Decision Letter 0]

19 Oct 2022

Dear Dr Maillard,

Thank you very much for submitting your manuscript "Early diagnosis of dengue: diagnostic utility of the SD BIOLINE Dengue Duo rapid test in Reunion Island" for consideration at PLOS Neglected Tropical Diseases. As with all papers reviewed by the journal, your manuscript was reviewed by members of the editorial board and by several independent reviewers. In light of the reviews (below this email), we would like to invite the resubmission of a significantly-revised version that takes into account the reviewers' comments. 

We cannot make any decision about publication until we have seen the revised manuscript and your response to the reviewers' comments. Your revised manuscript is also likely to be sent to reviewers for further evaluation.

Sincerely,

Puneet Bhatt, MD

Guest Editor

Elvina Viennet

Section Editor

Reviewer's Responses to Questions

**Key Review Criteria Required for Acceptance?**

**Methods**

-Are the objectives of the study clearly articulated with a clear testable hypothesis stated?

-Is the study design appropriate to address the stated objectives?

-Is the population clearly described and appropriate for the hypothesis being tested?

-Is the sample size sufficient to ensure adequate power to address the hypothesis being tested?

-Were correct statistical analysis used to support conclusions?

-Are there concerns about ethical or regulatory requirements being met?

Reviewer #1: 1. In the Flow chart Fig 1; Information regarding break up of RT-PCR positive in NS1Ag positive and IgM positive samples would be useful for the reason being IgM positive samples by RDT which were RT-PCR negative (being used as Reference standard) would have resulted inputs regarding any cross reactive antibodies to other closely related Flaviviruses such as West Nile or Japanese encephalitis viruses. This could have been resolved by more specific µ-capture IgM ELISA test in such samples.

2. Whether RT-PCR inhibitors were taken care of in the samples which were found negative by RT-PCR but were found positive by RDT for Dengue antibodies?

Reviewer #2: The authors made only patients record review only in this study. Is it possible to say retrospective study? 

What is the main topics of this manuscript. According to the title, we understood that the usefulness of RDT kits for diagnosis of dengue infection by comparing gold standard tests. But in this study, the authors did not use the full set of gold standard tests for confirmation of IgM Ab. The authors described that gold standard tests used RT-PCR. It is not full set because the authors also used IgM Ab for diagnosis of acute DENV infection. The authors should also test one gold standard IgM Ab tests which is validated for no cross reactivity with other flaviviruses. This is the biggest weak point of this study and the test performance is decreased due to not include IgM Ab at gold standard.

According to the flowchart Fig-1, it is very difficult to interpret for the readers. The authors described step by step but only 325 samples done for reference tests. So the sample size only 325. For calculation of the sensitivity, specificity, all sample must do reference tests to check your RDT test kit is correct or not? The author need to analyse only 325 samples and the authors must do gold standard IgM Ab ELISA test for publication to meet the standard at the reputed international journal.

Reviewer #3: Sample size calculation not mentioned

**Results**

-Does the analysis presented match the analysis plan?

-Are the results clearly and completely presented?

-Are the figures (Tables, Images) of sufficient quality for clarity?

Reviewer #1: 3. As seen from S3 table “Performance of NS1 and IgM RDT….”, column “4” on IgM RDT; sensitivity percentage on Day 1 being highest (65%) as compared to other subsequent days including more than 5 days (61%) where as my understanding is that Dengue IgM antibody rise in blood is after 3-5 days and can remain detectable till 90 days.

Reviewer #2: At Table-1. The authors described the demographic data and clinical features of study population. Concerning gender, we noted Male” population but cannot find for Female anywhere. Please revise it.

In table 1, RDT was tested on 671 patients. But according to case categories, total number is only 547. What are the discrepancies? And please check it.

As you mentioned that the sensitives were slightly better beyond 5th day, if possible please show the comparison of sensitivities based on day of illness

Reviewer #3: Yes

**Conclusions**

-Are the conclusions supported by the data presented?

-Are the limitations of analysis clearly described?

-Do the authors discuss how these data can be helpful to advance our understanding of the topic under study?

-Is public health relevance addressed?

Reviewer #1: NS1 Ag detection by RDT beyond 05 days is indicative of lasting viremia and could have been interesting to see its co-relation with viral load copy number by real time RT-PCR. It is interesting finding which could suggest changing viral dynamics. (For Reference: J Clin. Diagn. Res 2016 Apr; 10 (4):1-4) where authors have found NS1Ag positivity maximum on day 2-5 post onset of illness.

Reviewer #2: The authors described that the standard test as RT-PCR. Please kindly mention that is IT either quantitative or conventional Reverse transcription Polymerase Chain Reaction (RT-PCR)? Moreover, the author should described that it is either one step or two steps? Please also describe the vaidity of this PCR system for cross reactivity of the primers and other flaviviruses and within four serotypes of DENV infection.

 In this study, the authors made RT-PCR test Is there any information about the serotype of DENV circulating at this DENV season. It will be interesting to the scientific community and now it is no more new information for the peer group scientific community.

Reviewer #3: No

**Editorial and Data Presentation Modifications?**

Reviewer #1: nil

Reviewer #2: (No Response)

Reviewer #3: (No Response)

**Summary and General Comments**

Reviewer #1: 1. In the Flow chart Fig 1; Information regarding break up of RT-PCR positive in NS1Ag positive and IgM positive samples would be useful for the reason being IgM positive samples by RDT which were RT-PCR negative (being used as Reference standard) would have resulted inputs regarding any cross reactive antibodies to other closely related Flaviviruses such as West Nile or Japanese encephalitis viruses. This could have been resolved by more specific µ-capture IgM ELISA test in such samples.

2. Whether RT-PCR inhibitors were taken care of in the samples which were found negative by RT-PCR but were found positive by RDT for Dengue antibodies?

3. As seen from S3 table “Performance of NS1 and IgM RDT….”, column “4” on IgM RDT; sensitivity percentage on Day 1 being highest (65%) as compared to other subsequent days including more than 5 days (61%) where as my understanding is that Dengue IgM antibody rise in blood is after 3-5 days and can remain detectable till 90 days. 

4. Similarly, NS1 Ag detection by RDT beyond 05 days is indicative of lasting viremia and could have been interesting to see its co-relation with viral load copy number by real time RT-PCR. It is interesting finding which could suggest changing viral dynamics. (For Reference: J Clin. Diagn.Res 2016 Apr; 10 (4):1-4) where authors have found NS1Ag positivity maximum on day 2-5 post onset of illness.

Reviewer #2: (No Response)

Reviewer #3: (No Response)

PLOS authors have the option to publish the peer review history of their article (what does this mean?). If published, this will include your full peer review and any attached files.

Reviewer #1: Yes: Ajay Kumar Sahni

Reviewer #2: No

Reviewer #3: Yes: Kundan Tandel
---

## [Decision Letter · Decision Letter 1]

12 Feb 2023

Dear Dr Maillard,

Thank you very much for submitting your manuscript "Early diagnosis of dengue: diagnostic utility of the SD BIOLINE Dengue Duo rapid test in Reunion Island" for consideration at PLOS Neglected Tropical Diseases. As with all papers reviewed by the journal, your manuscript was reviewed by members of the editorial board and by several independent reviewers. The reviewers appreciated the attention to an important topic. Based on the reviews, we are likely to accept this manuscript for publication, providing that you modify the manuscript according to the review recommendations. 

Thank you for responding to reviewer's comments. 

Please note the few last comments attached. 

Thanks so much.

Kind regards,

Elvina Viennet

Sincerely,

Elvina Viennet, PhD

Section Editor

Elvina Viennet

Section Editor

Thank you for responding to reviewer's comments. 

Please note the few last comments attached. 

Thanks so much.

Kind regards,

Elvina Viennet

Reviewer's Responses to Questions

**Key Review Criteria Required for Acceptance?**

**Methods**

-Are the objectives of the study clearly articulated with a clear testable hypothesis stated?

-Is the study design appropriate to address the stated objectives?

-Is the population clearly described and appropriate for the hypothesis being tested?

-Is the sample size sufficient to ensure adequate power to address the hypothesis being tested?

-Were correct statistical analysis used to support conclusions?

-Are there concerns about ethical or regulatory requirements being met?

Reviewer #1: All queries resolved. Accept

Reviewer #2: (No Response)

Reviewer #4: (No Response)

**Results**

-Does the analysis presented match the analysis plan?

-Are the results clearly and completely presented?

-Are the figures (Tables, Images) of sufficient quality for clarity?

Reviewer #1: All queries resolved. Accept

Reviewer #2: (No Response)

Reviewer #4: (No Response)

**Conclusions**

-Are the conclusions supported by the data presented?

-Are the limitations of analysis clearly described?

-Do the authors discuss how these data can be helpful to advance our understanding of the topic under study?

-Is public health relevance addressed?

Reviewer #1: All queries resolved. Accept

Reviewer #2: (No Response)

Reviewer #4: (No Response)

**Editorial and Data Presentation Modifications?**

Reviewer #1: All queries resolved. Accept

Reviewer #2: Accept

Reviewer #4: (No Response)

**Summary and General Comments**

Reviewer #1: All queries resolved. Accept

Reviewer #2: In revision, the manuscript was improved. i have no more comments. I accepted as present version.

Reviewer #4: (No Response)

PLOS authors have the option to publish the peer review history of their article (what does this mean?). If published, this will include your full peer review and any attached files.

Reviewer #1: Yes: Ajay Kumar Sahni

Reviewer #2: No

Reviewer #4: No

Figure Files:

Data Requirements:

Reproducibility:

References

---

## [Editor Report · Decision Letter 2]

20 Mar 2023

Dear Dr Maillard,

We are pleased to inform you that your manuscript 'Early diagnosis of dengue: diagnostic utility of the SD BIOLINE Dengue Duo rapid test in Reunion Island' has been provisionally accepted for publication in PLOS Neglected Tropical Diseases.

Best regards,

Puneet Bhatt, MD

Guest Editor

Elvina Viennet

Section Editor

---

## [Editor Report · Acceptance letter]

27 Mar 2023

Dear Dr Maillard,

We are delighted to inform you that your manuscript, "Early diagnosis of dengue: diagnostic utility of the SD BIOLINE Dengue Duo rapid test in Reunion Island," has been formally accepted for publication in PLOS Neglected Tropical Diseases.

Best regards,

Shaden Kamhawi

co-Editor-in-Chief

Paul Brindley

co-Editor-in-Chief
